# Dental and Maxillofacial Cone Beam CT—High Number of Incidental Findings and Their Impact on Follow-Up and Therapy Management

**DOI:** 10.3390/diagnostics12051036

**Published:** 2022-04-20

**Authors:** Michael J. Braun, Thaddaeus Rauneker, Jens Dreyhaupt, Thomas K. Hoffmann, Ralph G. Luthardt, Bernd Schmitz, Florian Dammann, Meinrad Beer

**Affiliations:** 1Division of Neuroradiology, Department of Diagnostic and Interventional Radiology, University Hospital Ulm, 89081 Ulm, Germany; michael-1.braun@uni-ulm.de (M.J.B.); bernd.schmitz@uni-ulm.de (B.S.); 2Department of Diagnostic and Interventional Radiology, University Hospital Ulm, 89081 Ulm, Germany; 3Department of Dentistry, University Hospital Ulm, 89081 Ulm, Germany; thaddaeus.rauneker@gmail.com (T.R.); ralph.luthardt@uniklinik-ulm.de (R.G.L.); 4Institute of Epidemiology and Medical Biometry, University Ulm, 89081 Ulm, Germany; jens.dreyhaupt@uni-ulm.de; 5Department of ENT, University Hospital Ulm, 89081 Ulm, Germany; thomas.hoffmann@uniklinik-ulm.de; 6i2SouI—Innovative Imaging in Surgical Oncology Ulm, University Hospital of Ulm, 89081 Ulm, Germany; 7Comprehensive Cancer Center Ulm (CCCU), University Hospital of Ulm, 89081 Ulm, Germany; 8Universitätsinstitut für Diagnostische, Interventionelle und Pädiatrische Radiologie (DIPR), University Hospital Bern, 3010 Bern, Switzerland; fd-radiologie@gmx.de; 9Center for Translational Imaging “From Molecule to Man” (MoMan), University Hospital of Ulm, 89081 Ulm, Germany

**Keywords:** CBCT, incidental findings, impact on therapeutic management

## Abstract

Cone beam computed tomography (CBCT) is increasingly used for dental and maxillofacial imaging. The occurrence of incidental findings has been reported, but clinical implications of these findings remain unclear. The study’s aim was to identify the frequency and clinical impact of incidental findings in CBCT. A total of 374 consecutive CBCT examinations of a 3 year period were retrospectively evaluated for the presence, kind, and clinical relevance of incidental findings. In a subgroup of 54 patients, therapeutic consequences of CBCT incidental findings were queried from the referring physicians. A total of 974 incidental findings were detected, involving 78.6% of all CBCT, hence 2.6 incidental findings per CBCT. Of these, 38.6% were classified to require treatment, with an additional 25.2% requiring follow-up. Incidental findings included dental pathologies in 55.3%, pathologies of the paranasal sinuses and airways in 29.2%, osseous pathologies in 14.9% of all CBCT, and findings in the soft tissue or TMJ in few cases. Clinically relevant dental incidental findings were detected significantly more frequently in CBCT for implant planning compared to other indications (60.7% vs. 43.2%, *p* < 0.01), and in CBCT with an FOV ≥ 100 mm compared to an FOV < 100 mm (54.7% vs. 40.0%, *p* < 0.01). Similar results were obtained for paranasal incidental findings. In a subgroup analysis, 29 of 54 patients showed incidental findings which were previously unknown, and the findings changed therapeutical management in 19 patients (35%). The results of our study highlighted the importance of a meticulous analysis of the entire FOV of CBCT for incidental findings, which showed clinical relevance in more than one in three patients. Due to a high number of clinically relevant incidental findings especially in CBCT for implant planning, an FOV of 100 × 100 mm covering both the mandible and the maxilla was concluded to be recommendable for this indication.

## 1. Introduction

Cone beam computed tomography (CBCT) was introduced for dental imaging by Piero Mozzo et al. in 1998 [1]. Rapidly, CBCT has found its way into clinical practice due to its superior spatial resolution for high-contrast structures such as bones and teeth [2]. Until recently, former standard techniques such as the orthopantogram (OPG) have been increasingly replaced by CBCT [3]. Besides the application in endodontics, periodontics, and orthodontics [4,5,6] CBCT has also demonstrated its importance in implant planning [7]. For this indication, a large field of view (FOV) is often necessary. In many cases, the entire mandible or the entire maxilla has to be depicted. Dief et al. conducted a systematic review of the literature regarding incidental findings in CBCT scans [8]. In seven out of ten reviewed articles, CBCT examinations were performed with a “large” FOV [9,10,11,12,13,14,15]. In three of these studies, the FOV was 130 × 130 mm^2^ or larger [9,12,14]. This imposes requirements on the imaging and the diagnostic process with regard to: 1. radiation exposure of the patient, 2. evaluation of a possibly large and complex three-dimensional examination area, and 3. interpretation of pathologies beyond the professional focus of the clinical specialist, especially for incidental findings (IF) and their implications for further therapy. Lopes et al. [16] detected an average of 4.72 incidental findings in a total of 47 examinations that covered both the mandible and the maxilla. Nguyen et al. found an average of 1.85 IF per scan in 555 patients of an older population for pre-implant assessment [17]. In this context, the aim of this study was the systematic analysis of a large consecutive patient group with regard to the frequency and especially the clinical impact of incidental findings depending on the indication for CBCT imaging and the size of the FOV.

## 2. Materials and Methods

Approval for this study was obtained from the Institutional Review Board and informed consent was obtained from each patient. First, 374 consecutive CBCT studies that were performed in a period of 33 months in a tertiary hospital were retrospectively reviewed and evaluated for technical parameters and pathological incidental findings. Follow-up imaging was excluded to avoid distortion of the results by therapies that had taken place in the meantime. In case of multiple examinations in one patient, the oldest examination of this patient was analyzed. In a second step, 54 patients out of the 374 patients that were included in the study were reviewed to assess the clinical relevance of the registered incidental findings.

### 2.1. Technical Examination Parameters and Standard Operating Procedure

All studies were acquired on a Morita 3D Accuitomo 170 CBCT (J. Morita Corp., Kyoto, Japan). This CBCT device consists of a solid frame construction with a 360° rotatable flat-panel detector. The patient was positioned on a customizable examination chair with fixations for the head and the chin to reduce moving artifacts. After performing two scout images, the FOV was selected out of nine available settings, ranging from 40 × 40 mm^2^ to 120 × 170 mm^2^. The voxel sizes ranging from 80 to 260 µm could be selected, depending on the chosen FOV. The indication of the CBCT examination was checked for every patient by a radiologist with expertise. In the vast majority of all cases, an FOV of 100 × 100 mm^2^ (275 patients, 73%) was selected. A voxel size of 250 µm was chosen in 82.4% (308 patients) of all cases for image reconstruction. In summary, 309 examinations were performed with an FOV of 100 × 100 mm^2^ or larger and a voxel size of 250 µm or more, whereas 65 patients were examined with an FOV < 100 × 100 mm^2^ and a voxel size of less than 250 µm, respectively (Table 1a,b).

The average amperage was 6.5 mA (SD 1.3 mA) with a maximum of 10 mA and a minimum of 3 mA. The average dose length product (DLP) was 116.43 mGy × cm (SD 28.03 mGy × cm) with a maximum of 246 mGy × cm and a minimum of 52 mGy × cm, wherein 50% of the values were between 87 and 140 mGy × cm (Table 2). The software i-dixel, offered by the manufacturer Morita, was used for the digital analysis of the three-dimensional datasets including multiplanar, dental, and 3D volume rendering reconstructions. 

### 2.2. Evaluation of the Results

Each CBCT examination was assessed with regard to their justifying indication, based on the guideline of the European Commission concerning “Cone Beam CT for Dental and Maxillofacial Radiology” [18]. Multiple primary indications were possible. The referring physicians were unbundled into their disciplines. Two radiologists (20 years and 5 years of experience in maxillofacial radiology at the time of data acquisition) evaluated all CBCT examinations in a consensus regarding incidental findings. The findings were split up into the following categories: 1. dental, 2. osseous, 3. soft tissue, 4. temporomandibular joints (TMJ), and 5. paranasal sinus airways. This classification system was chosen in accordance with the German ICD-10 [19]. In a consensus evaluation in cooperation with a board certified dentist, the incidental pathologies were classified with regard to their presumed therapeutical relevance according to a three-step scale: findings were rated as “red” if therapy was supposed to be necessary, “yellow” if follow-up was presumed to be sufficient, and “green” if there was no further treatment required. 

### 2.3. Subgroup Analysis

In a second part of the study, a subgroup of 54 patients was selected, who were send to CBCT by the four main referring physicians with long-lasting professional experience. The referring physicians were interviewed by questionnaire on the following statements: 1. did the CBCT examination provide a new diagnosis and/or a new result? (YES or NO) and 2. has there been any change in therapeutic management affected due to a new diagnosis or new result? (YES or NO)?

### 2.4. Statistical Methods

Continuous data were reported as mean, standard deviation, and min-max. Ordinal and categorical data were analyzed as absolute and relative frequencies. Additionally, pie charts were used to visualize the distribution of categorical data. For group comparisons of categorical data, the chi square test or Fisher’s exact test was used as appropriate. Logistic regression analysis was used to determine associations of FOV ≥ 100 × 100 mm^2^, implantologic indication, radiation exposure, and dental findings. A two-sided *p*-value ≤ 0.05 was considered to be statistically significant. Because of the explorative nature of this study, all results from statistical tests had to be interpreted as hypothesis-generating and not confirmatory. An adjustment for multiple testing was not made. Statistical analysis was performed with SAS, version 9.4 (SAS Institute, Cary, NC, USA).

## 3. Results

In total, 374 patients including 209 women and 165 men were examined. Patients were 8–90 years old with an average age of 50.9 (±22.3) years.

### 3.1. Referring Physicians and Primary Indications

The patients were referred by a total of 59 physicians, including 46 (78%) dentists and 13 (22%) non-dentists, and almost the half of the latter was composed of otolaryngologists. 

About two thirds of all patients (252 patients, 67.4%) were referred by dentists without further specialization followed by orthodontists (50 patients, 13.4%) and otolaryngologists (28 patients, 7.5%) (Table 3a). 

The primary indication for CBCT included implant planning in more than half of the cases (n = 191, 51.1%), followed by orthodontics (n = 56, 15%) and ENT pathologies (n = 36, 9.4%) (Table 3b).

### 3.2. Pathological Findings

In total, 1601 pathological findings were detected among all patients, which was 4.3 pathologies per patient. Out of these, 974 (60.8%) findings were categorized as incidental, involving 78.6% of all patients. These incidental findings included 353 (36.2%) green-, 245 (25.2%) yellow-, and 376 (38.6%) red-rated pathologies. Unbundled to their etiologies, 539 (55.3%) findings originated in dental pathologies, 284 (29.2%) of the findings were located in the paranasal sinuses and airways, and 145 (14.9%) were related to osseous tissue. Four (0.4%) and two (0.2%) pathologies were located in the paramaxillofacial soft tissue and the TMJ, respectively.

### 3.3. Dental Findings 

In total, 292 red-, 240 yellow-, and 245 green-rated dental incidental findings were registered. More than 50% of all patients had at least one red-rated dental pathology. Subgroup analysis of red pathologies revealed a significant influence of the FOV: when the FOV was <100 × 100 mm^2^, 40.0% of patients showed at least one incidental finding compared to 54.7% of all patients who were examined with an FOV ≥ 100 × 100 mm^2^ (*p* = 0.05). There was no significant difference in the detection of incidental findings between examinations with implantologic indication (77.0%) vs. all other indications (84.2%) (*p* = 0.08). However, 60.7% of the patients with an implantologic indication had red-rated findings whereas less than the half (43.2%) of the other group had red-rated incidental dental findings (*p* < 0.01). Multiple logistic regression analysis showed significant influence of the implantologic indication with more detected relevant IF. Odds ratio for red IF was 1.88 (95% CI: 1.22 to 2.88). Figure 1 shows an illustrative case of periapical disease.

### 3.4. Paranasal Sinuses and Airways Findings 

In total, 140 red-, 71 yellow-, and 173 green-rated incidental findings were detected in the paranasal sinuses and airways. Of all patients, 32% had at least one red-rated pathology of the paranasal sinuses and airways. The detection rate of incidental findings in total was more than doubled when using an FOV ≥ 100 × 100 mm^2^ compared to an FOV < 100 × 100 mm^2^ (63.4% vs. 29.2%, *p* < 0.01). When focusing on red-rated findings, the difference was even more pronounced: almost three times more incidental findings were registered when using an FOV ≥ 100 × 100 mm^2^ compared to an FOV < 100 × 100 mm^2^ (36.9% vs 12.3% *p* < 0.01) (Appendix A). The primary indication “implantology” for CBCT had a lower but still statistically significant influence on the detection rate of red-rated incidental findings. Multiple logistic regression analysis showed similar results (Odds ratio 5.54 (95% CI: 2.47 to 12.43), *p* < 0.01), favoring the FOV ≥ 100 × 100 mm^2^ with more detected relevant IF (Appendix A). Figure 2 shows an illustrative case of a maxillary sinusitis, probably in the context of a fungal infection.

### 3.5. Osseous Findings

In total, 33 red-, 54 yellow-, and 342 green-rated osseous incidental findings were detected. About 8% of all patients had at least one red-rated osseous pathology. Significantly more osseous incidental findings were registered when applying an FOV ≥ 100 × 100 mm^2^ compared to an FOV < 100 × 100 mm^2^ overall (73.8% vs. 46.2%, *p* < 0.01). When focusing on red-rated incidental findings, the total number of these findings was low, and the differences were not significant when comparing FOV ≥ 100 × 100 mm^2^ vs. FOV < 100 × 100 mm^2^ (9.1% vs.: 4.6%, *p* = 0.62). The primary indication for CBCT examination had significant influence on the difference in the frequency detection of osseous incidental findings: among examinations with implantologic indication, 98.4% of CBCT revealed osseous incidental findings compared to only 38.3% on all other indications (*p* < 0.01). Again, when focusing on red-rated findings, the difference between both indication groups was statistically not significant (8.9% vs. 7.7%, *p* = 0.85). Multiple logistic regression analysis showed no significant influence of both, indication implantology and FOV, on relevant IF. Odds ratio was 1.04 (95% CI: 0.49 to 2.23), *p* = 0.91) for indication implantology and 1.93 (95% CI: 0.55 to 6.80) for FOV, respectively. Figure 3 shows two illustrative cases of an osteoma and an osteoblastic metastasis.

### 3.6. Soft Tissue Findings

No green-rated and only three yellow-rated soft tissue incidental findings were registered. Merely one red-rated finding of an ill-defined enlarged submandibular lymph node was detected. There was no significant difference in the frequency of incidental soft tissue findings regarding implantologic indications or FOV (*p* = 1.00).

### 3.7. TMJ Findings

Incidental pathologic findings of the TMJ were registered in only five patients. Clinical relevance was classified as “yellow” in all of these cases. There was significant influence of indication regarding implantologic indications (0% vs. 1.3%, *p* = 0.03). All five relevant yellow IF were found in patients without an implantologic indication.

### 3.8. Clinical Subgroup Analysis

The extended clinical analysis of 54 selected patients showed the following results: in more than half of all these patients (n = 29, 53.7%), 98 incidental diagnoses were newly detected by the CBCT examination, including 25 green-, 19 yellow-, and 54 red-rated pathologies. Within the red-rated findings, dental pathologies were most frequent with 75.9% (n = 41) followed by pathologies of the paranasal sinuses with 18.5% (n = 10) and osseous pathologies with 5.6% (n = 3). 

In 19 patients (35.2%), the therapy management was altered due to new diagnoses. In these particular patients, a total of 63 pathological findings were detected, including 37 (58.7%) red-rated pathologies. Within these red-rated findings, there were 25 patients with periradicular disease (67.6%) and 5 (13.5%) with pathologies of the paranasal sinuses. Seven patients had other pathologies (18.9%) (Figure 4).

## 4. Discussion

In accordance with guidelines [18,19] and other studies [20,21,22] regarding a routine patient group, the vast majority of our patients were admitted to three-dimensional imaging with CBCT due to planning implant placement. Before inserting dental implants, adequate imaging is demanded and CBCT therefore seems to be the method of choice [23,24]. 

More than 80% of the examinations that were included in our study were performed with an FOV of 100 × 100 mm^2^ or larger. This size of FOV reliably covers the entire maxillary and mandibular dental arch and allows the evaluation of the adjacent structures of the midface that are potentially relevant for the planning of dental therapies. Alareddy et al. published the actual largest number of cases focusing on incidental findings in dental imaging, with an FOV of 130 × 130 mm [9]. Edwards et al. also applied a large but not exactly quantified field of view “…from the roof of the orbits inferiorly to at least the second cervical vertebrae” [25]. Other authors described the FOV as “large” [10,11,13,15]. Price et al. applied FOV of 150 × 150 mm^2^ up to 220 × 220 mm^2^. Our mainly used FOV with 100 × 100 mm^2^ was rather small compared to the studies mentioned above. In some other studies, a smaller FOV was chosen [16,26,27]. The non-homogenously distributed number of examinations with a specific FOV was certainly a limitation of our study. The results of Alareddy et al. showed an incidence of 4.3 pathological findings per examination in all patients, whether they were included in the primary indication or not. Edwards et al. reported an incidence of 1.97 incidental finding per examination [25]. A pathological finding is often considered as “incidental” if it is not in context with the primary indication [10,11,12,28,29]. Moreover, incidental findings with reference to the primary indication might be regarded as incidental, especially if they are asymptomatic. Alareddy et al. documented 943 incidental pathologies in 1000 patients, also not distinguishing if they were inside or outside the region of interest [9]. 

We detected 2.6 incidental findings per patient with 78.6% of all patients showing incidental findings. In the study of Price et al., 272 CBCT scans revealed 881 incidental findings, equivalent to 3.2 per scan [12]. Caglayan and Tozoglu estimated the overall rate of incidental findings as 92.8% in a group of 207 consecutive patients [20]. Warhekar et al. described only 7.2% incidental findings in a cohort of 795 consecutive patients [30], which stands in contrast to the much higher rate of incidental findings in our study as well as most other studies [16,20,21,22,26,29]. A possible explanation is that Warhekar et al. analyzed written reports of CBCT examinations instead of performing a systematic image analysis by themselves. 

Published reports differ significantly concerning the type and localization of incidental findings. Price et al. [12] as well as Caglayan and Tozoglu [10] described pathologies of the airways as the most common incidental finding, comprising 35% and 51.8% of all incidental findings, respectively. In our study, pathologies of the paranasal sinuses and airways comprised only 14.9% of incidental findings. In contrast, dental pathologies were the most frequent incidental findings in our study with 55.3%, which only comprised 11.3% and 26% of the incidental findings in the above-mentioned studies, respectively. 

Differences between published studies were also found concerning the frequency of indication for a CBCT examination. In contrast to our and most other studies, Caglayan and Tozoglu included only 15 out of 207 patients for implant planning [10]. 

Despite differences in study concepts, composition of patient groups, or indications, our results are broadly in line with other published data and demonstrated a high frequency of incidental pathological findings in CBCT of the maxillofacial region, whether they were in the region of interest or not [20,21,22,26]. 

The high incidence of 80% of dental incidental findings in all patients outlines the clinical importance of a meticulous image analysis. More than the half of all findings were classified as clinically relevant. “Red” clinically relevant dental as well as airway incidental findings occurred in 61% of CBCT for implant planning, and thus almost 50% more frequently in patients who were admitted to CBCT for implant planning compared to CBCT for other indications. This might be explained by a generally lower sanitary dental status and a higher incidence of paranasal sinusitis in patients who need dental implants [31,32]. Considering the significantly reduced detection rate of dental incidental findings in CBCT using an FOV < 100 mm, our findings emphasize the application of an FOV of 100 mm × 100 mm^2^, covering both the mandible and the maxilla in the context of an implant planning situation. 

In nearly every second patient of our subgroup with extended clinical analysis, a new diagnosis was found that was not known before CBCT examination. In two thirds of patients with these new detected diagnoses, the therapeutic management had to be adjusted. This is different to the results of Lopes et al., where most of the detected IF were classified not to undergo further treatment or referral to another professional [22]. Our subgroup analysis confirmed the reduced number of therapeutically relevant findings when using an FOV < 100 × 100 mm^2^, especially when regarding dental incidental findings as well as implantologic indications for CBCT. It could be assumed that at least some of these incidental findings may have been missed if the FOV was limited too close to the site of primary clinical interest [26]. Possible therapeutic complications or even implant failure may result in individual inconveniences for the patients and also monetary consequences for the public health system. This can only be estimated and should be a goal of further studies. 

Radiologists must deal with a holistic diagnostic work-up covering a clinically reasonable area that is not obligatorily limited to the scope of the referring specialist. This work-up includes especially the detection and description of incidental findings. Radiologists should deal and familiarize with the specific analysis of CBCT to minimize the possible consequences for the patients of missing incidental findings. This emphasizes again the importance of close collaborations between medical and dental specialties as Khalifa et al. recently pointed out [29].

## 5. Conclusions

CBCT of the maxillofacial region revealed a high percentage of clinically relevant additional findings. This study presents data underlining the clinical relevance of these findings. Our results confirmed the influence and dependency of the findings on the FOV and the primary indication, especially for implant planning. The “incidental” findings induced a change of therapy in more than one in three patients. Due to a high number of clinically relevant incidental findings in CBCT for implant planning, an FOV of 100 x 100 mm was concluded to be recommendable for this indication. A meticulous analysis of the entire FOV is essential.

## Figures and Tables

**Figure 1 diagnostics-12-01036-f001:**
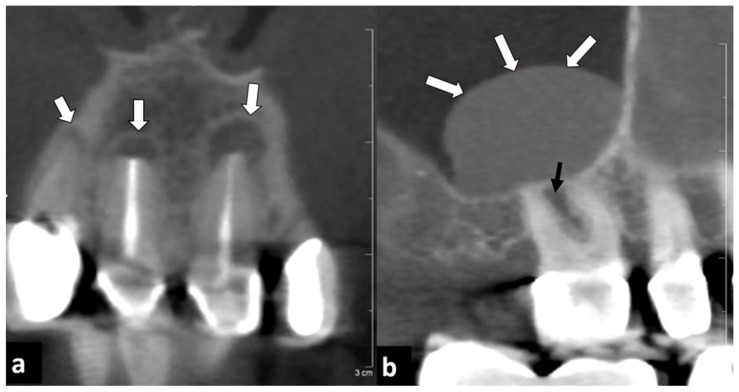
Forty-one year old male patient who underwent endodontal therapy. (**a**) Paracoronar reconstruction showed periapical resorptions and osteolysis at teeth 21, 11, and less pronounced also at tooth 12 (arrows). (**b**) Parasagittal reconstruction of the upper right jaw showed a polypoid mucosal swelling at the bottom of the right maxillar sinus (white arrows), probably induced by an interradicular resorption at tooth 16 (black arrow).

**Figure 2 diagnostics-12-01036-f002:**
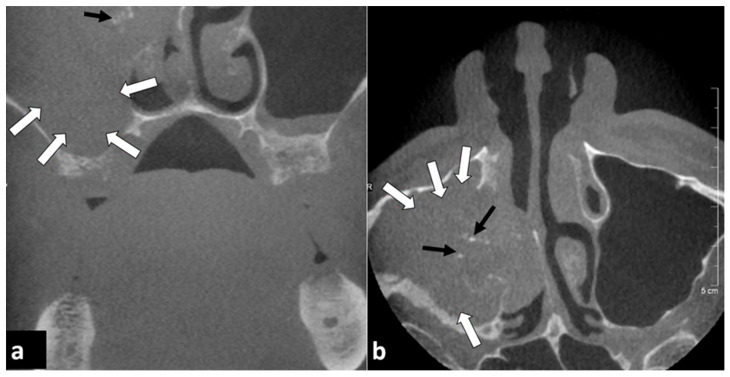
Eighty-three year old male patient. CBCT was performed for implant planning. (**a**) Coronal and (**b**) axial reconstructions showed a maxillary sinusitis on the right side with mucosal swelling (white arrows) bulging into the nasal cavity. Sparse calcifications (black arrows) within the mucosal swelling may indicate a fungal infection.

**Figure 3 diagnostics-12-01036-f003:**
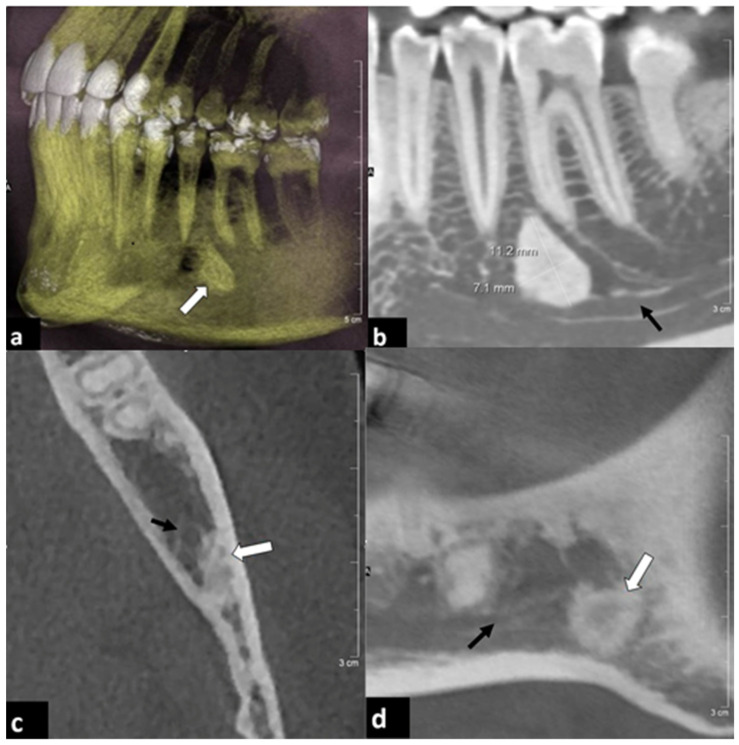
Two examples of osseous incidental findings. (**a**,**b**) Forty-four year old female patient. CBCT was performed prior to bisphosphonate medication due to osteoporosis. (**a**) Volume rendering and (**b**) parasagittal reconstruction showed an osteoma (white arrow) between the dental roots of teeth 35 and 36. The alveolar nerve canal (black arrow) was likely to be narrowed. (**c**,**d**) Seventy-five year old female patient with history of breast cancer and numbness of the lower left jaw. (**c**) Paraxial and (**d**) parasagittal reconstructions showed a sclerotic lesion (white arrow) with blurred contact to the alveolar nerve (black arrow). Further clinical evaluation revealed the diagnosis of an osteoblastic bone metastasis.

**Figure 4 diagnostics-12-01036-f004:**
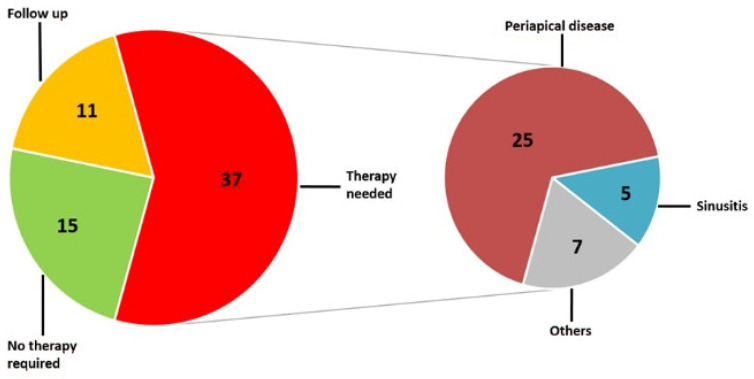
Pie chart of all incidental findings among those patients whose therapeutic management had to be adjusted due to the findings in CBCT and DVT. Presented is a breakdown of the red-rated incidental findings to their etiologic origin.

**Table 1 diagnostics-12-01036-t001:** (**a**) Number and frequency of applied FOV. (**b**) Number and frequency of voxel sizes used for image reformation.

**(a)**	
**FOV in mm^2^**	**Number of Examinations**
100 × 100	275
80 × 80	44
140 × 140	18
60 × 60	14
170 × 170	13
40 × 40	5
Other	5
**(b)**	
**Voxel Size in μm.**	**Number of Examinations**
250	308
160	44
125	16
80	5
260	1

**Table 2 diagnostics-12-01036-t002:** Frequency of radiation exposition for CBCT examinations.

Exposition, DLP (mGy × cm)	Number of Examinations
52	1
70	13
72	1
87	97
105	41
122	127
140	63
154	3
157	9
175	15
215	1
246	3

**Table 3 diagnostics-12-01036-t003:** (**a**) Subspecialization of referring physicians. (**b**) Frequency of primary indications for CBCT examinations.

**(a)**	
**Referring Physicians**	**Number of Referred Patients**
General dentist	252
Orthodontic dentist	50
Otolaryngologists	28
Others	44
**(b)**	
**Primary Indication**	**Number of Examinations**
implantology	191
orthodonctics	56
ENT	36
endodontics	21
teeth removal	18
oncology	13
trauma	8
TMJ	3
parodontics	1
other	27

## Data Availability

Data supporting reported results can be found in the Hospital’s PACS system.

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
