# Peer review of "Dental and Maxillofacial Cone Beam CT—High Number of Incidental Findings and Their Impact on Follow-Up and Therapy Management"

_diagnostics, 2022, doi:10.3390/diagnostics12051036_

Round 1

Reviewer 1 Report

The study is interesting, but some steps need to be better described. It can be better illustrated the expertise of the operators, for example, in terms of working experience in years and specialization. 
Guidelines used for diagnosis are indicated, but the reference or a link has to be reported in the manuscript.
The same is for the German ICD- 10 classification system. It is not a national journal, so a reference or a link in English has to be indicated or the classification summarized in a table.
The limitations of the study are not adequately described.
For example, It’s not clear which type of examination was used to diagnose before referring the patient for CBCT.
Was it the same for all the patients? 
The Physicians referred the patients with a complete medical report? Most of them were dentists. In general, the dentist is very concise in requesting a radiographic examination. So is difficult to assess whether the CBCT changes the diagnosis.
It is also not so objective that two radiologists and one dentist can define what needs further therapy or if  follow-up can be sufficient
These issues have to be evaluated in the discussion. 

Author Response

Point-to-Point response to reviews

We thank all reviewers for their helpful comments. We largely agree with the points raised and considered them in the revised version of the manuscript. In the following, our changes are listed next to the points raised.

Reviewer #1 (anonymous)

Point 1:It can be better illustrated the expertise of the operators, for example, in terms of working experience in years and specialization. “

Reply:  Thanks for the notice.

Radiologist #1 is a professor in radiology and board certified Radiologist. He was head of department at the hospital at the time of data collection. He has a working experience of more than 20 years with subspecialisation in dentomaxillar, facial as well as head and neck radiology.

Radiologist #2 is a board certified consultant in radiology and neuroradiology. He worked as a full time resident for radiology in the department of radiology at the hospital of data collection. He had a working experience of five years in radiology with a focus on dentomaxillar imaging for two years at the time of data acquisition.

The participating board proofed dentist had a working experience of one year at the time of data acquisition. The study was part of his doctoral thesis.

We illustrated the working experiences in the manuscript.

Point 2: „Guidelines used for diagnosis are indicated, but the reference or a link has to be reported in the manuscript. The same is for the German ICD- 10 classification system. It is not a national journal, so a reference or a link in English has to be indicated or the classification summarized in a table. “

Reply: Thank you very much for the notice.

 we added the links to the german ICD-10 classification system and the European guideline for radiation protection No 172 to the references [4, 5].

Point 3: „It’s not clear which type of examination was used to diagnose before referring the patient for CBCT. Was it the same for all the patients? “

Reply: Thank you very much for the remark.

The patients, who we included consecutively in this study were part of the clinical routine process in the department of radiology. For this reason, the preceding examination of the patients depended on the subspecialisation of the referral physician. However, we assume with a high degree of certainty that the CBCT became necessary as a consequence of a correct clinical examination.

Point 4: „The Physicians referred the patients with a complete medical report? “

Reply: As mentioned in point 3, all patients were part of the clinical routine process. So, all patients were referred with an adequate clinical report to determine the justifying indication for the radiological examination of cone beam CT. A complete medical report was therefore not available for all patients. For the selectively examined patients in our subgroup analysis, however, a complete review of the medical files was carried out in consultation with the treating dentist or physician, respectively.

Point 5: „In general, the dentist is very concise in requesting a radiographic examination. So is difficult to assess whether the CBCT changes the diagnosis. It is also not so objective that two radiologists and one dentist can define what needs further therapy or if follow-up can be sufficient. These issues have to be evaluated in the discussion. “

Reply: The assignement of “treatment changing CBCT findings” was made on basis of the interviews with the referring physicians for a subgroup of 54 patients, and was achieved by consensus expertise of the operators for the remaining patients. 

Reviewer 2 Report

Given the "high N" - lots patients placed into groups, I do not understand wht a true stiatistical analysis was not performed were p values and trues\ biologic stats would be helpful to prove the point of the study and establish validity

Use X2 or another statistic techniqe for statistical soundness. 

Author Response

Point-to-Point response to reviews

We thank all reviewers for their helpful comments. We largely agree with the points raised and considered them in the revised version of the manuscript. In the following, our changes are listed next to the points raised.

Reviever #2 (anonymous)

Point 1: „Given the "high N" - lots patients have been placed into groups, I do not understand why a true statistical analysis wasn´t performed (missing p values). Trues\ biologic stats would be helpful to prove the point of the study and establish validity. Use X2 or another statistic technique for statistical soundness. “

Reply: Thanks for the remark. In the revised version of the manuscript, we did further analysis of group:
1) according to field of view (FoV; FoV <100 mm vs. FoV ≥100 mm)

2) according to voxel size (<250 µm vs. ≥250 µm

3) Indication implantology (yes vs. no)

Group comparisons were performed for of categorical data only. The chi square test or Fishers exact test was used as appropriate.

Reviewer 3 Report

Dear Authors,

thank you for this article... however, there are many many articles out there on incidental findings, and in the introduction you must point out what is your key research questions, and why it is innovative or different, or even similar...

In stead of discribing why CBCT is usefull, we all know this now, it is more than 20 years on the market, more interesting would be to explain which articles already have researched incidental findings en what is missinf or to be questioned.

Furthermore, no explanation on you r methodology is given: why this methodology, and is it similar or why is it different to other articles..

also in the discussion: include ALL literature up to today, there are many more interesting articles out there

thanks

Author Response

Point-to-Point response to reviews

We thank all reviewers for their helpful comments. We largely agree with the points raised and considered them in the revised version of the manuscript. In the following, our changes are listed next to the points raised.

Reviever #3 (anonymous)

Point 1: „…in the introduction you must point out what is your key research questions, and why it is innovative or different, or even similar. “

Reply: Thanks a lot for the advice!

Our main research questions were as follows:

  1. Do incidental findings in implantological indications have a clinical significance?
  2. Do a large FoV and a larger voxel size have a significant impact on the detection of clinically relevant incidental findings?

As in several previous studies on the subject, we mainly have impantological questions. This should make our study comparable with current studies.

Since we used a large FoV in most cases due to the clinical issues, there is an important difference to other studies on the subject in this regard. In many studies, the FoV has been limited (Lopes et al. 2016; Johnson et al. 2021).

Point 2: „Instead of describing why CBCT is useful, we all know this now, it is more than 20 years on the market, more interesting would be to explain which articles already have researched incidental findings and what is missing or to be questioned. “

Reply: Thank you very much for this advice.

We added recent studies to our references, that examined the impact of incidental findings in CBCT of the maxillofacial region and put them in context with our results.  We could not confirm the statement of Warhekar et al. [25] that orofacial malignancies were the most common detected incidental findings.

Point 3: „Furthermore, no explanation on your methodology is given: why this methodology, and is it similar or why is it different to other articles. “

Reply: Thank you for the question.

The main difference in our method lies in the application of the large FoV and large Voxel size in the context of routine clinical examinations. The area of investigation in other studies was mostly limited to the pathology to be investigated or a small FoV. (Lopes et al. 2016, Johnson et al. 2021, Oser et al. 2017).

The main similarity with other studies in this field is the spectrum of indications with a focus on implantology. This makes our study comparable to other studies with a quite similar volume of data.

Point 4: „…include ALL literature up to today, there are many more interesting articles out there. “

Reply:  Thanks for the remark.

To the best of our knowledge, we added all recent literature concerning incidental findings in dentomaxillofacial cone beam CT imaging. We referred to them in the discussion and linked the articles in the references.

Round 2

Reviewer 3 Report

Dear authors,

again the introduction has not been changed: you have to indeed state your main purpose of review, but as mentioned before, describe why this is different from existing articles on incidental findings on CBCT, rather than describing CBCT technology and use

your results are not clearly organized... you should have tables/explanations with which findings exactly and so on...

but especially you emphasize your study method on voxel and field of view, and the results are not showing clear influence of this

statistical analysis is required for multiple variables, not just descriptives...

the discussion is totally drawing WRONG conclusions

how can one recommend larger fields for implantology if you do not look at ALL Variables:

first of all the field of view and the JAW TYPE (mandible, singe implant, versus full arch maxilla is totally different), and the also the VOXEL etc, and the associated dose!

please do not make suggestions and insinuations if they are out of context

it is not because an incidental finding by questionnaire changed therapy, that we know what changed, and especially what clinical info was looked at and known.. and also was it relevant and would it influence the final primary goal outcome????

thorough revisions are needed

Author Response

Point-to-Point response to review

We again thank reviewer 3 for her/his helpful comments. We largely agree with the points raised and considered them in the revised version of the manuscript. In the following, our changes and argumentations are listed next to the points raised.

Reviewer 3 (anonymous)

Point 1: “again the introduction has not been changed: you have to indeed state your main purpose of review, but as mentioned before, describe why this is different from existing articles on incidental findings on CBCT, rather than describing CBCT technology and use.“

Reply: Thanks again for the advice! We changed our focus in the introduction text to the clinical impact of implantologic indications and the FOV (p. 2, line 69-72; 77-86). We regard this as the main purpose of our study and also the main difference to other studies. As requested we also deleted parts of CBCT-technology descriptions, which we assumed to be standard knowledge meanwhile (p. 2, line 59-66).

Point 2: “your results are not clearly organized... you should have tables/explanations with which findings exactly and so on...“

Reply: Thank you very much for this advice. We added tables for radiation exposition (Tab.2, p. 4, line 125) and referring physicians (Tab. 3a, p.5, line 172). As suggested, we also added the results of our multiple logistic regression analysis with regard to Field of View, implantologic indications and radiation exposure (Tab.4, suppl.). In the supplement tables we also added the frequency of YELLOW and GREEN findings for all anatomical locations.

Point 3: but especially you emphasize your study method on voxel and field of view, and the results are not showing clear influence of this

Reply: Thank you for the remark. In our univariate analysis, there were more clinically relevant incidental dental findings in a FOV ≥ 100 x 100 mm² with borderline significance for RED findings (p=0.05). Indeed, our results in the multiple logistic regression analysis showed no statistical influence of these variable. But the FOV still had statistical influence on relevant incidental airway pathologies (RED), favouring a larger FOV to detect more clinically relevant (RED) incidental findings. Since the voxel size is strongly dependent on the chosen FOV, we did not consider it as an independent factor of influence in our multiple logistic regression analysis.

Point 4: statistical analysis is required for multiple variables, not just descriptives…“

Reply:  Thanks for the remark. We did a multiple logistic regression analysis on all concerning influential variables (FOV ≥ 100 x 100 mm² (yes or no), Indication Implantology (yes or no)) and radiation exposition (DLP, continuous variable). We added the results to the previous, univariate analysis. Odds Ratio and p-values are reported in the text and supplemental (p. 6-8; Tab 4, suppl.).

Point 5: “the discussion is totally drawing WRONG conclusions

how can one recommend larger fields for implantology if you do not look at ALL Variables:

first of all the field of view and the JAW TYPE (mandible, singe implant, versus full arch maxilla is totally different), and the also the VOXEL etc, and the associated dose!

please do not make suggestions and insinuations if they are out of context

it is not because an incidental finding by questionnaire changed therapy, that we know what changed, and especially what clinical info was looked at and known.. and also was it relevant and would it influence the final primary goal outcome????

Reply: Thank you for the comment. We tried to accommodate your objection. Therefore, we did the additional multiple logistic regression analysis with regard to the influence of the larger FOV, the implantologic indication and the radiation exposition.

Because voxel size is technically closely related to the chosen FOV, we do not consider it as an independent variable.  As recommended, we included the radiation exposition in the calculation of the other variable.

With both the univariate and the multivariate analysis, we were able to determine a significant influence of the FOV and the implantological indication to clinically relevant (RED) incidental findings. Indeed, the multivariate analysis showed no significant influence of the larger FOV for RED dental findings with p=0.128. But our univariate analysis showed a minor but nevertheless significant influence with a p-value of 0.05. In both statistical analyses, the influence of a larger FOV on the detection of clinically relevant incidental findings of the airways was clearly given (p < 0.01).

The significant influence of the implantation indication was clearly given in the univariate analysis and also an influencer in the multivariate statistics.

In the vast majority of our examinations, we covered both the maxilla and the mandible with the FOV of 100 x 100 mm in one examination. Therefore, no dedicated selection of one part of the jaw had to be made.

In summary, based on our data, we believe that the recommendation for at least a FOV of 100 x 100 mm² to detect relevant incidental findings in implantologic regards is nonetheless correct.